# The Role of an Acidic Peptide in Controlling the Oxidation Process of Walnut Oil

**DOI:** 10.3390/foods8100499

**Published:** 2019-10-15

**Authors:** Yu Jie, Hongfei Zhao, Bolin Zhang

**Affiliations:** Beijing Key Laboratory of Forest Food Processing and Safety, Beijing Forestry University, No. 35 Qinghua East Road Haidian District, Beijing 100083, China; 13020021532@163.com (Y.J.); zhaohf518@163.com (H.Z.)

**Keywords:** lipid oxidation, antioxidant peptide, acidic amino acid residues

## Abstract

Here, the mechanism of action of an antioxidant peptide rich in acidic amino acid residues in controlling lipid oxidation is discussed. Firstly, in the presence of this peptide, the fluorescence intensity of lipid peroxide in samples of walnut oil was very low, indicating that the peptide prevented the formation of lipid peroxides. Secondly, the production of lipid-derived radicals of oil was reduced by 23% following addition of the anti-oxidative peptide. Thirdly, Raman shifts of the lipid with the anti-oxidative peptide showed that acidic amino acid residues of the peptide were involved in delaying lipid oxidation. Finally, seven peptide inhibitors were synthesized with variations to the amino acid sequence of the original peptide, and Glu–Asp was proven to enhance the peptide’s superoxide anion radical scavenging activity and decrease the formation of linoleic acid peroxides. Our findings emphasize the potential value of acidic amino acid residues in protecting unsaturated fatty acids from oxidation.

## 1. Introduction

Edible oils and fats are indispensable to the human diet. Unsaturated fatty acids (UFAs) exert many health benefits such as prevention of heart disease, hypertension, and diabetes [1,2,3]. Recently, there has been a rapid increase in the production of edible oils or fats enriched with selected UFAs [4]. However, these oils or fats often undergo autoxidation, eventually causing loss of flavor and nutritional value. Additionally, lipid peroxidation has negative effects on proteins, enzymes, and membrane structure, leading to damage to organs or tissues in the human body [5,6]. Thus, to protect UFA-enriched products, oxidation of these lipids must be decelerated. 

Lipid oxidation is well-known to be a chain reaction induced by free radicals. In the initiation step, a lipid radical is formed by cleavage of a carbon bond or by transferring an initiator free radical to the hydrogen atom of a carbon bond. In the propagation step, molecular oxygen reacts with a lipid radical and then transfers a peroxyl radical to a hydrogen atom of the carbon bond to generate another carbon radical [7]. Therefore, once the oxygen-centered radicals or lipid-derived radicals are bound by a radical quencher, the chain reaction will slow down. Peptides are known to be free radical quenchers. Many extracted peptides show high resistance to linoleic acid oxidation, and the reason why these peptides can resist oxidation was attributed to their hydrophobic amino acid residues [8,9,10,11]. However, Saiga, Tanabe and Nishimura [12] proved that acidic amino acid residues in peptides were the key to reducing the oxidation of lipids. To date, no more studies have been done that explore the possibility of using peptides rich in acidic amino acid residues to retard lipid oxidation. Our previous work discovered the very promising antioxidant QITEGEDGGG, an acidic peptide that was effective in delaying the oxidation of walnut oil. It was found that after 30 days of storage, the peroxide value of the peptide-added walnut oil was only 1.2% higher than that of commercial antioxidant tertiary butyl hydroquinone (TBHQ)-added walnut oil, and 30% lower than the peroxide value of antioxidant-free oil. We suspected that the acidic amino acid residues or glycine residues in the peptide acted to increase the antioxidant activity [13]. However, whether the presence of acidic amino acid residues enhances the antioxidant activity of the peptide remains incompletely understood.

In this study, we aimed to determine the mechanism by which the acidic peptide QITEGEDGGG inhibits the oxidation of lipids and test the effects of acidic amino acid residues of the peptide on inhibiting lipid oxidation. Specifically, the interaction between the peptide and lipid was evaluated, the lipid-radical formed during oxidation was monitored and characterized, and the ability of acidic amino acid residues of the peptide to inhibit lipid oxidation and scavenge oxygen radicals was investigated.

## 2. Materials and Methods

### 2.1. Materials

Seven peptides QITEGEDGGG, ITEGEDGGG, QITEGED, QEGEDGGG, QITEGGGG, HIQKEDVPSER, and HIQKVPSER were synthesized by SciLight Biotechnology Ltd. (Beijing, China) based on the literature [13,14]. Virgin walnut oil was provided by Xinmu Biotechnology Ltd. (Beijing, China). Linoleic acid, linolenic acid, arachidonic acid and oleic acid were purchased from J&K Chemical (Hebei, China). Nt-butyl-α-phenylnitrone (PBN) was obtained from Sigma-Aldrich (St. Louis, MO, USA). C11-BODIPY 581/591 was purchased from Thermo Fisher Scientific (Waltham, MA, USA) and fluorescein isothiocyanate (FITC) was purchased from Lablead Biotech (Beijing, China).

### 2.2. The Mechanism of Action of Acidic Peptide in Oil

#### 2.2.1. Preparation of Oil Samples

According to EU regulations that limit the concentration of antioxidants in oil, 0.02 g/L of the peptide QITEGEDGGG was added to walnut oil as the experimental sample. Walnut oil without any antioxidants was the control.

#### 2.2.2. Observation of Lipid Peroxide

To clarify the action of the peptide QITEGEDGGG in inhibiting oxidation of walnut oil, the distribution of peptide and peroxides from oxidized oil droplets was observed. The inhibition of the formation of lipid peroxide by the peptide was monitored by confocal laser scanning microscopy (Leica SP8, Wetzlar, Germany). Both the experimental and control samples were stored at 37 °C for 24 h for oxidation. The peroxides of the experimental sample and control were dyed by C11-BODIPY 581/591, which is an ideal probe of lipid oxidation [3]. The peptide of the experimental sample was marked by FITC. The excitation wavelengths of C11-BODIPY 581/591 and FITC were at 568 nm and 495 nm, respectively. Emission of C11-BODIPY 581/591 and FITC were monitored over the range of 570 to 640 nm (red channel) and 500 to 545 nm (green channel), respectively. The fluorescence intensity of peptide and peroxide was used to characterize the ability of the peptide to inhibit lipid oxidation. 

#### 2.2.3. Evaluation of Lipid-Derived Radical Production

The lipid-derived carbon-centered radicals of oxidized oil were studied to compare walnut oil oxidation taking place in the presence the peptide QITEGEDGGG with that taking place in the absence of the peptide. A JEOL FA-200 electron spin resonance (ESR) spectrometer (JEOL Ltd., Tokyo, Japan) was used to determine the intensity of lipid-derived radicals of the experimental and control samples during oxidation. PBN was used as an ESR spin trapper to capture the lipid-derived carbon-centered radicals in oxidized walnut oil [15]. First, 10 mg PBN was dissolved in a 10 mL walnut oil sample to obtain a PBN–walnut oil mixture (1 mg/mL). Next, 0.02 g/L of peptide was added to another PBN–walnut oil mixture. Two PBN–walnut oil mixtures were heated for oxidation. ESR spectra were measured every hour during the heating process until the appearance of the radical signal. Additionally, 10 mg PBN was added to a peptide aqueous solution and treated under the same conditions. The ESR spectrometer settings were microwave power, 0.998 mW; center field, 322.657 mT; sweep width, 5 mT; sweep time, 4 min.

#### 2.2.4. Examination of the Lipid-Derived Radical

Gas Chromatography and Mass Spectrometry (GC-MS) was employed to identify the type of lipid-derived radicals detected by the ESR spectrometer. The Shimadzu (Kyoto, Japan) GC-MS QP2010 ultra system was equipped with a Shimadzu AOC-20i auto injector and Rxi-5Sil MS fused silica capillary column (30 m × 0.25 mm I.D., film thickness 0.25 μm, 5% phenyl/95% dimethylpolysiloxane, Restek, Bellefonte, PA, USA). Helium was used as the carrier gas at a flow rate of 1.0 mL/min, and the injector port temperature was 250 °C in split mode with a ratio of 1:10. The GC oven temperature was initially held at 35 °C for 5 min and then increased to 320 °C at a rate of 10 °C/min for 5 min. The mass spectrometer was operated in the electron impact (EI) ionization mode at an ion source temperature of 200 °C. The GC-MS interface temperature was maintained at 250 °C and the electron energy was kept at 70 eV. 

#### 2.2.5. Investigation of the Raman Shift of the Lipid

To observe walnut oil changes before and after oxidation, the Raman shifts of the peptide QITEGEDGGG and UFAs in oil droplets were recorded (LabRAM HR Evolution, Horiba Jobin Yvon S.A.S., Longjumeau, France). Virgin walnut oil and walnut oil containing 0.02 g/L of peptide QITEGEDGGG were heated to accelerate oxidation. Raman spectra were measured every hour during heating until the Raman signal showed. Additionally, a peptide dry powder was also treated with the same conditions to evaluate the Raman shift of the peptide. The parameters were as follows: excitation light source of 532 nm for average scan time of 15 s, grating of 600 lines mm^−1^, and a sweep range of 200–2000 cm^−1^.

Linoleic acid, oleic acid, linolenic acid, and arachidonic acid are the four major UFAs in walnut oil [16,17]. Therefore, the Raman shifts of these four UFAs with or without the addition of the peptide were also examined as they underwent the process of oxidation. The spectra were acquired using the following conditions: the excitation light source of 785 nm, exposure time 30 s for 20 exposures, grating of 600 lines mm^−1^, and a sweep range of 200–2000 cm^−1^.

### 2.3. Effect of Acidic Amino Acid Residues in Anti-Oxidative Peptides

#### 2.3.1. Superoxide Radical Scavenging Assay

To probe the acidic amino acid residues’ activity in the anti-oxidative peptide QITEGEDGGG, the antioxidant activities of seven synthesized peptides (QITEGEDGGG, ITEGEDGGG, QITEGED, QEGEDGGG, QITEGGGG, HIQKEDVPSER, and HIQKVPSER) were measured by a superoxide radical scavenging assay. The following reaction system was used. First, the seven synthesized peptides were dissolved in deionized water. Then, 0.30 mL of each peptide solution was mixed with 0.75 mL of dihydronicotinamide adenine dinucleotide (936 µM), 0.75 mL of nitro blue tetrazolium (300 µM), and 0.75 mL of phenazine methosulfate (120 µM) [13]. The solutions were incubated for 5 min and then the absorbance was read at 560 nm. The superoxide anion-scavenging activities of peptide inhibitors were calculated as follows:
Superoxide anion scavenging-ability (%) = (1 − (Asample/Acontrol)) × 100%(1)
where Acontrol was the absorbance without sample and Asample was the absorbance with sample.

#### 2.3.2. Linoleic Acid Peroxidation Inhibition Assay

The ability of the seven synthesized peptides (QITEGEDGGG, ITEGEDGGG, QITEGED, QEGEDGGG, QITEGGGG, HIQKEDVPSER, and HIQKVPSER) to inhibit the oxidation of linoleic acid was evaluated as described by Jie et al. [13]. Linoleic acid and Tween 20 (1:1 *v/v*) were mixed in phosphate buffer (0.2 M, pH 7.0) to obtain a linoleic acid emulsion. Each peptide solution was combined with an equivalent volume of linoleic acid emulsion and incubated in the dark at 37 °C for 24 h, enabling oxidation. The degree of linoleic acid oxidation was determined from the ferric thiocyanate reaction. To assay 100 µL of oxidative mixture, 4.7 mL of ethanol (75%), 100 µL of ammonium thiocyanate (3.94 M), and 100 µL of ferrous chloride (0.02 M) solution in HCl (0.10 M) were added. The absorbance of the solution was read at 500 nm after 3 min of the reaction. The lipid peroxidation inhibition activity was calculated using the following equation:
Lipid peroxidation inhibition-ability (%) = (1 − (Asample/Acontrol)) × 100%(2)
where Acontrol was the absorbance without sample and Asample was the absorbance with sample.

### 2.4. Statistical Analysis

The superoxide radical scavenging assays and linoleic acid oxidation inhibition assays were carried out in triplicate and the values represented as the mean ± standard deviation (SD). Analysis of variance (ANOVA) with Tukey comparison was performed to measure the significance of differences and (*p* < 0.05).

## 3. Results

### 3.1. The Mechanism of Action of an Acidic Peptide in Oil

#### 3.1.1. Reduction of Lipid Peroxide

The effects of the peptide QITEGEDGGG on lipid oxidation were examined by confocal laser scanning microscopy. The fluorescence signal of peroxide in oil droplets of walnut oil which underwent oxidation is displayed in Figure 1a. Figure 1b is the fluorescence signal of peroxide in oil with the peptide added (peptide-oil) and Figure 1c presents the fluorescence signal of the peptide in the peptide-oil mix. Figure 1d represents the summary of the results from Figure 1a, Figure 1b,c. It is a comparison of peroxide level in peptide-free oil and peptide-oil mix, as well as the peptide level and peroxide level in peptide-oil mix. Peroxide level and peptide level are expressed as the mean value of intensity of fluorescence signals (A.U.). The mean value of the fluorescence intensity of peroxide in peptide-free oil was 988 A.U., while the mean value of the fluorescence intensity of peroxide in peptide-oil mix was 47 A.U. It can be seen that the production of peroxide in peptide-oil mix was reduced a lot. Furthermore, the fluorescence intensity of the peptide in the same oil droplet was also higher than that of the peroxide. Thus, data from Figure 1d shows that the intensity of the fluorescence signal from the peroxide was reduced after the peptide was added to the walnut oil. 

#### 3.1.2. Inhibition of Lipid-Derived Radical Production

To investigate the lipid-derived radical or carbon-based radical scavenging activity of the peptide against lipid peroxidation, the radicals were trapped using PBN and detected by ESR. The ESR spectra of both oxidized walnut oil (OWO) and oxidized peptide–oil (OPO) were two typical triplet spectra (Figure 2a). The triplet spectra revealed a carbon-based radical trapped by PBN, which appeared as a composite signal. Figure 2b shows that no radical is trapped in the heat-treated peptide solution or in un-oxidized walnut oil. Therefore, the peptide produced no carbon-centered radicals during lipid oxidation, and thus the lipid-derived radicals were all produced by walnut oil. The intensity of carbon-centered radicals was reduced by 23% in OPO compared with that in OWO.

GC-MS was used to identify the radicals. The results are the same as those obtained by ESR. Only one PBN adduct was detected by GC-MS (t_R_ = 20.06 min) (Figure 2c). In addition, the substance (t_R_ = 10.16 and 10.95 min) may be a degradation product of PBN. It was formed by the cleavage of the tert-butyl group from the PBN moiety and tentatively identified as benzaldehyde. 

#### 3.1.3. Peptide Binding with UFAs

The peptide scavenged lipid-radicals in OWO, indicating that the peptide may react with walnut oil. Thus, the interaction between the peptide and walnut oil was examined by Raman microscopy. The Raman spectra of the un-oxidized oil, OWO and OPO in the range of 800–2000 cm^−1^ are shown in Figure 3a. The major peaks were observed at Raman bands of 1746 cm^−1^ (C=O stretching), 1658 cm^−1^ (*cis*-C=C stretching), 1439 cm^−1^ (C–H deformation), 1301 cm^−1^ (C–H bending), 1264 cm^−1^ (=C–H bending), 1080 cm^−1^ (C–C stretching), 974 cm^−1^ (*trans*-C=C bending), and 872 cm^−1^ (C–C stretching). No other peaks were generated in OPO and the intensity of the OPO remained unchanged, showing the same main peaks as un-oxidized oil. In contrast, the intensity of OWO was increased in all spectra.

The peptide played a role in preventing lipid oxidation, and its molecular conformation was unchanged after heating (Figure 3b). However, the Raman bands of the heat-treated peptide were enhanced at all shifts, particularly at 867 cm^−1^ (C–O–C stretching, O–O stretching), 1032 cm^−1^ (C–C stretching), 1249 cm^−1^ (amide III, C–N stretching, and N–H stretching), 1255 cm^−1^ (amide III, C–N stretching, and N–H stretching), 1424 cm^−1^ (CH_2_ deformation and CH_3_ deformation), 1675 cm^−1^ (amide I, C=O stretching, N–H bending, and C–N stretching). Comparison of Figure 3a,b shows that the spectra of both peptide and walnut oil increased in intensity during oxidation, such as the bands at approximately 800–1000, 1200, 1400, and 1600 cm^−1^. These shifts were associated with C–C stretching, C–H deformation, C–N stretching, and C–O stretching. 

The Raman spectra of four UFAs with or without the addition of the peptide are shown in Figure 3c–f. Generally, the position of the main characteristic peaks of the Raman spectra of the four UFAs were the same as those of walnut oil. The Raman band at 1654 cm^−1^ stands for *cis*-C=C stretching, the region of 1439–1454 cm^−1^ stands for C–H deformation, the band 1301 cm^−1^ stands for C–H bending, the band 1264 cm^−1^ stands for =C–H bending, and the band 1080 cm^−1^ stands for C–C stretching. The shifts of the main peaks of the four UFAs with the peptide were lower than the spectral shifts of the four UFAs without it.

### 3.2. Effect of Acidic Amino Acid Residues in Peptides

In a previous study, the peptide QITEGEDGGG inhibited the production of superoxide anion radicals and linoleic acid oxidation by 86.46% and 60.37%, respectively (Inhibitor No. 0 in Table 1) [13].

In this study, to verify whether acidic amino acid residues contribute to the antioxidant capacity of the peptide QITEGEDGGG, the amino acid residues in the peptide were removed systematically. Firstly, the amino acid residues at the beginning and end of the decapeptide were replaced in inhibitor No. 1 (ITEGEDGGG) and inhibitor No. 2 (QITEGED), respectively. The inhibitory activities of these two peptides were not significantly different from that of inhibitor No. 0 (*p* > 0.05). Thus, the amino acid residues at the beginning or end of the peptide did not affect their activity. Next, inhibitor No. 3 (QEGEDGGG) was synthesized. The loss of the Ile and Thr amino acid residues had no effect on the ability of the peptide to scavenge superoxide anion radicals, and to inhibit lipid peroxidation.

Lastly, Glu–Asp was removed from inhibitor No. 0 QITEGEDGGG and inhibitor No. 4 (QITEGGGG) was synthesized. Compared with inhibitor No. 0, the ability of inhibitor No. 4 to scavenge superoxide anion radicals and inhibit linoleic acid oxidation was significantly decreased (*p* < 0.05). Inhibitor No. 4 was only half as effective as inhibitor No. 0 in inhibiting oxidation of linoleic acids. Clearly, the presence of Glu–Asp affects the peptide’s antioxidant activity. To check the efficacy of dipeptide Glu–Asp, the inhibitor No. 5 (ED) was synthesized. However, the antioxidant activity of dipeptide Glu–Asp was much lower than that present in the decapeptide.

By screening the amino acids in the decapeptide, it was determined that the acidic amino acids Glu–Asp are the active amino acid residues that confer the antioxidant activity of the peptide. To provide a comparison with the decapeptide and to begin to assess the generality of these observations, we drew upon the literature to synthesize a hydrophobic peptide containing Glu–Asp (HIQKEDVPSER inhibitor No. 6) and then synthesized a version of that peptide that eliminated the Glu–Asp amino acid residues (HIQKVPSER inhibitor No. 7) [14]. The superoxide anion radical scavenging activity of inhibitor No. 6 was significantly enhanced compared to that of inhibitor No. 7 (*p* < 0.05). The ability of inhibitor No. 7 to inhibit oxidation of linoleic acid was significantly decreased by 22% compared to that of inhibitor No. 6 (*p* < 0.05).

## 4. Discussion

Previous studies suggested that anti-oxidative peptides containing numerous hydrophobic amino acids or amino acids with OH groups could increase the oxidation resistance of linoleic acids [18,19,20,21]. The role of acidic amino acid residues has been mentioned, but few studies have been done to explore their significance.

In this work, we investigated the anti-oxidative properties of an acidic peptide and found that the acidic amino acid residues enhanced retardation of walnut oil oxidation. The walnut oil used in this test is comprised of 12,15-octadecadienoic acid (55.13%), 9-octadecenoic acid (20.32%), hexadecanoic acid (9.96%), stearate (8.68%), 9,12-octadecadienoic acid (1.30%), and 11-eicosenoic acid (1.21%). Other antioxidant components like tocopherol in walnut oil are present in minute quantities (0.11%), and have negligible effect on controlling the oxidation process of walnut oil. Therefore, the added peptide QITEGEDGGG contributed to the retardation of oxidation of walnut oil. 

Figure 1 and Figure 2 demonstrated that the peptide QITEGEDGGG reduced the generation of lipid peroxides and the production of lipid-radicals. Although the ESR spectra did not exhibit hyperfine splitting into doublets, it can be concluded that the PBN radical adducts detected in our results were a mixture of carbon-based radicals by comparing our results with the literature using PBN or POBN as spin to trap lipid-radicals [15,22,23,24,25,26,27,28]. The PBN radical adducts are mixtures because of the viscosity of the oil, which results in decreased mobility of the radicals. The same phenomenon was also found by Thomsen, Kristensen and Skibsted [22] who observed triplets of doublet signals in rapeseed oil by ESR, but the doublets were not observed in all spectra.

Do acidic amino acid residues play a role in the above two aspects? Raman shifts of the un-oxidized walnut oil, OWO, OPO and the peptide proved that the acidic amino acid residues of peptide QITEGEDGGG were involved in retarding the lipid oxidation. The Raman bands at 1658 cm^−1^ and 974 cm^−1^ represented the cis-double bonds and trans-double bonds of PUFAs, respectively [29,30,31]. In case of OWO, the expansion of the stretching vibration of cis-double bonds represented cis-fatty acid fragmentation. In contrast, the expansion of deformation of trans-double bonds indicated that the level of trans-fatty acids in walnut oil grew rapidly. C–C stretching occurred in the region from 872 to 1080 cm^−1^, indicating the fragmentation of fatty acids and thus generation of more carbon-based radicals to promote the chain reaction. The intensity increase of the band at 1746 cm^−1^ in the OWO represented a gradual break of C=O in fatty acids, thus producing more fragmentation of the chain that enabled the peroxyl radical to be carried [7]. Contrarily, in the case of OPO, no increase in trans-fatty acids level and subsequent chain fragmentation was observed (Figure 3a). Likewise, the *cis*-C=C stretching, C–H deformation, C–H bending, =C–H bending, and C–C stretching of the four UFAs in the presence of the peptide were slightly reduced (Figure 3c–f). We conclude that the peptide retards the fracturing of the C–H bonds of UFAs.

It was noted that an increased intensity in the Raman shift of the peptide took place during oil oxidation (Figure 3b). The region between 1249 and 1255 cm^−1^ in the peptide spectra corresponded to the amide III band [32]. The increased intensity of the amide III band in the heat-treated peptide indicated alterations in the β-sheet of the peptide. The region of 800–950 cm^−1^ or 1500–1800 cm^−1^ represented the Raman spectra of acidic amino acid residues and their amides [33]. The band shift at 856 cm^−1^ was because of the C=O stretching of the Gln residue. In addition, the band at 1670 cm^−1^ in the spectra of the peptide represents the amide I band stretching of Glu residues and the band at 1438 cm^−1^ is associated with the Asp residue. The stretching of acidic amino acid residues causes their side chains to be exposed, improving their radical scavenging ability and enhancing the anti-oxidative ability of the peptide. Apparently, the Glu and Asp residues of the peptide are involved in retarding the generation of cleaved C–H bonds of the UFAs. In addition, hydrogen bonding was formed between the peptide and the fatty acids from walnut oil because more overlap between the peptide and UFAs occurred in the Raman spectra. Our data showed that both =C–H bending and N–H stretching appeared in the same region (Figure 3). Stretching of the *cis*-C=C of fatty acids and the shift in amide I of the Glu residue were observed in the same region. 

Subsequently, we determined that the combination of Glu–Asp improved the antioxidant activity of peptides by screening the amino acids residues in the original decapeptide and synthesizing Glu–Asp-containing and Glu–Asp-missing peptide inhibitors (Table 1). In the absence of Glu–Asp, both superoxide anion radical scavenging activity and lipid anti-oxidative activity of peptide inhibitors were reduced. Acidic amino acid residues were reported to chelate metal ions to prevent lipid peroxidation [12,34]. However, in our previous studies (data not shown), the decapeptide, which was rich in acidic amino acid residues, failed to enhance the metal-chelating activity. In this study, the acidic amino acid combination Glu–Asp increased superoxide anion radical scavenging activity and decreased the generation of lipid peroxides. This suggests that acidic amino acid residues may limit the effects of metal ions by scavenging superoxide anions, and thereby obstructing the generation of H_2_O_2_, which promotes the oxidation of metal ions. Furthermore, the presence of acidic amino acid resides in peptides prevented the generation of lipid peroxides and the reaction of oxygen-centered radicals and lipid-derived radicals to form peroxyl radicals (Figure 4). 

Based on our results, we speculate that a peptide would have high antioxidant activity if it has acidic amino acid residues. 

## 5. Conclusions

The acidic peptide can reduce the oxidation of lipids, and this is most likely due to the acidic amino acid residues. Acidic amino acid residues were shown to eliminate oxygen-centered radicals and formed hydrogen bonds with UFAs. The combination of Glu–Asp in peptides has a positive effect on superoxide anion scavenging and inhibits oxidation of linoleic acid. Thus, we believe that acidic amino acid residues may reduce the lipid oxidation by their oxygen-centered radical scavenging activity and by decreasing formation of lipid peroxides. Our findings suggest that peptides containing Glu–Asp may guide development of further anti-oxidative peptides, which confer resistance to oxidation of UFA-enriched products.

## Figures and Tables

**Figure 1 foods-08-00499-f001:**
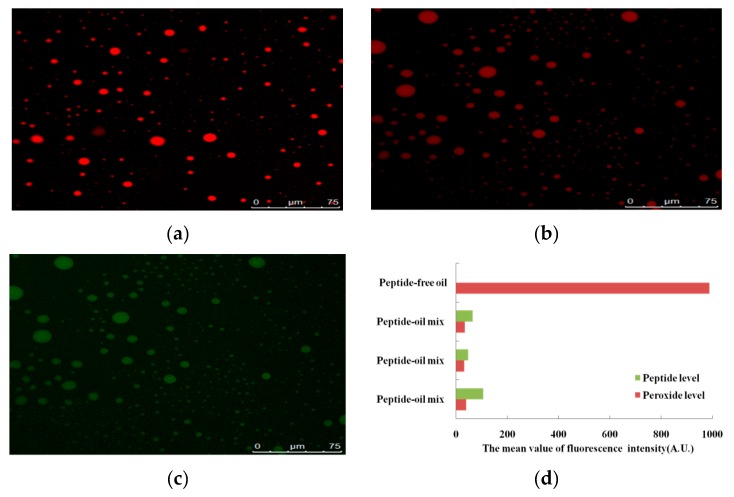
Effect of the peptide QITEGEDGGG against peroxides in oxidation oil. (**a**) The distribution of peroxide in walnut oil without any antioxidant. (**b**) The distribution of peroxide in peptide–oil. (**c**)The distribution of peptide in peptide–oil. (**d**) A comparison of the peroxide level in peptide-free oil and peptide-oil mix, as well as the peroxide level and peptide level in peptide-oil mix. Peroxide level and peptide level are expressed as the mean value of intensity of fluorescence signals (A.U.). Red fluorescence presents the signal intensity of C11-BODIPY, the dye associated with peroxides, while the green fluorescence is the signal intensity of FITC, the dye associated with the peptide. Scale bar is 75 μm.

**Figure 2 foods-08-00499-f002:**
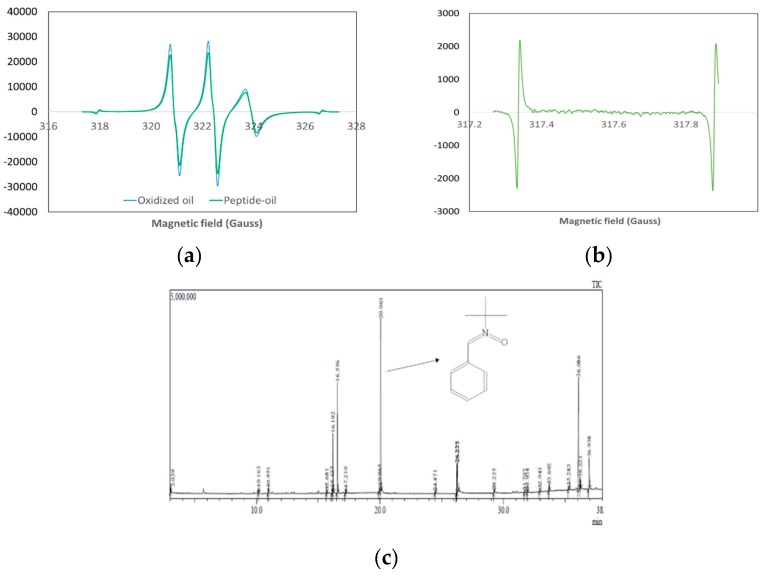
Effect of the peptide QITEGEDGGG against lipid radicals in oxidation oil. (**a**) ESR spectra of the PBN-radical adduct of walnut oil and peptide–oil during oxidation. (**b**) ESR signal of the heat-treated peptide solution. (**c**) GC-MS of the PBN-radical adduct in oxidized oil.

**Figure 3 foods-08-00499-f003:**
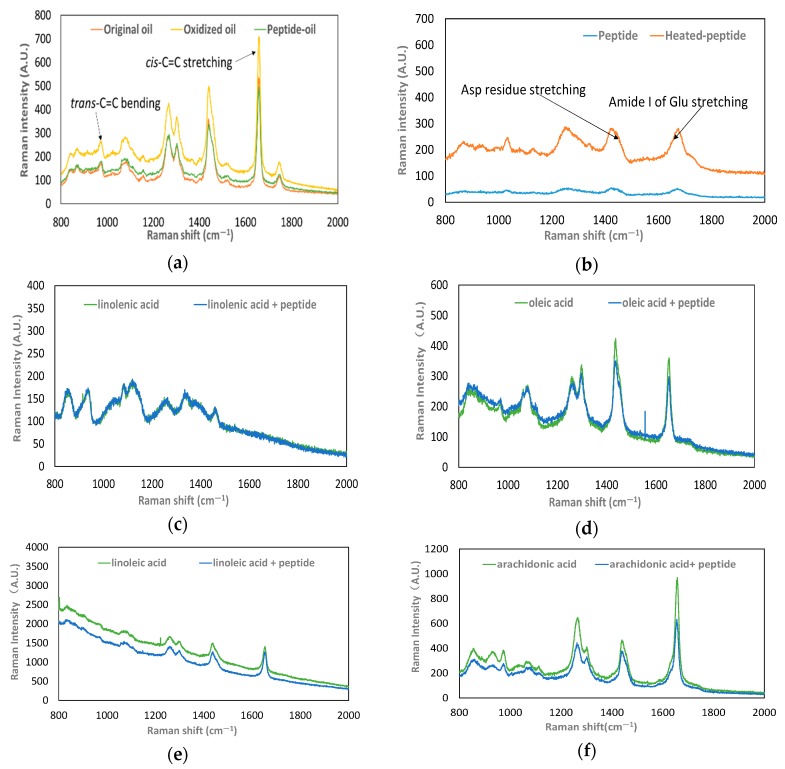
Raman spectra of lipid in the range from 800 cm^−1^ to 2000 cm^−1^. (**a**) Raman spectra of un-oxidized walnut oil, oxidized walnut oil, and oxidized peptide–oil. (**b**) Raman spectra of the peptide and the heat-treated peptide. (**c**) Raman spectra of oxidized linolenic acid and oxidized linolenic acid with peptide. (**d**) Raman spectra of oxidized oleic acid and oxidized oleic acid with peptide. (**e**) Raman spectra of oxidized linoleic acid and oxidized linoleic acid with peptide. (**f**) Raman spectra of oxidized arachidonic acid and oxidized arachidonic acid with peptide.

**Figure 4 foods-08-00499-f004:**
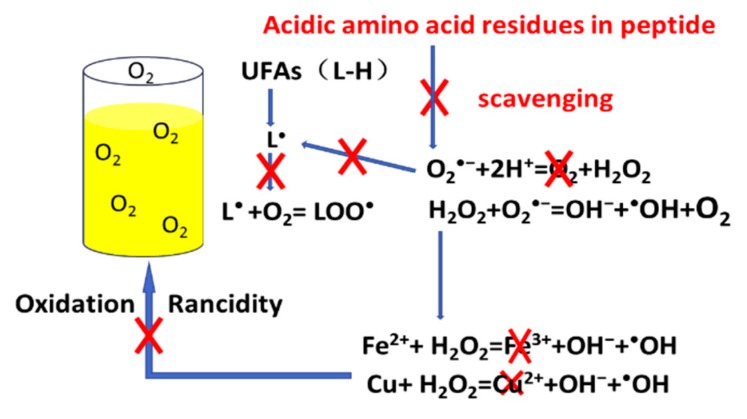
The possible mechanism of acidic amino acids in the peptide to resist oil oxidation. L-H stands for the unsaturated fatty acids in oil, L• represents the carbon-base radicals and LOO• represents the peroxyl radicals.

**Table 1 foods-08-00499-t001:** The inhibitory activity of peptide inhibitors against superoxide anion radical and lipid oxidation.

Inhibitor No.	Sequence	Inhibition of Superoxide Anion Radical (%)	Inhibition of Linoleic Acid Oxidation (%)	Reference
0	QITEGEDGGG	86.46 ± 0.07 ^a^	60.37 ± 0.16 ^a^	[13]
1	ITEGEDGGG	85.03 ± 0.05 ^a^	58.62 ± 0.39 ^a^	
2	QITEGED	85.00 ± 0.07 ^a^	55.34 ± 0.39 ^a^	
3	QEGEDGGG	85.54 ± 0.23 ^a^	53.49 ± 0.35 ^a^	
4	QITEGGGG	64.14 ± 0.02 ^b^	31.11 ± 0.17 ^b^	
5	ED	14.08 ± 0.05 ^c^	27.28 ± 0.15 ^c^	
		**control**		
6	HIQKEDVPSER	87.28 ± 0.01 ^d^	52.53 ± 0.13 ^d^	[14]
7	HIQKVPSER	62.36 ± 0.02 ^e^	30.50 ± 0.16 ^e^	

Values are given as the means ± SD from triplicate determinations. a–e means in the same column with different letters differ significantly (*p* < 0.05).

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
