# Peer review of "The Role of an Acidic Peptide in Controlling the Oxidation Process of Walnut Oil"

_foods, 2019, doi:10.3390/foods8100499_

Round 1

Reviewer 1 Report

The manuscript entitled “The role of an antioxidant peptide on controlling the oxidation process of walnut oil”, authored by Yu Jie, Hongfei Zhao and Bolin Zhang, investigates the role of acidic amino acid residues in antioxidant properties of short peptides and suggests the potential role of such peptides as antioxidants for natural oil preservation. I find the study quite interesting, however a few thoughts have crossed my mind whilst reading it.

Major remarks

The study lacks a valuable comparison with any antioxidant used in the food industry for oil preservation, and particularly with the most common one, which is alpha-tocopherol. Keeping in mind the limit on antioxidant amount in oil, mentioned in the line 62, it would be desirable to compare antioxidant potential of the studied peptides, or at least a representative one, with alpha-tocopherol within the same molar as well as mass concentration.

Another question to be asked is why did the authors not examine the dipeptide Glu-Asp alone? There are reports demonstrating that different dipeptides have antioxidant porperties. Why not to check Glu-Asp in this study?

The last thing which bothers me is the figure 1 and the interpretation of the results. The panels (b) and (c) the oil droplets are much bigger than those in the panel (a). Are the graphics scaled properly? If so, is it possible that addition of the peptide alters the surface tension, which subsequently leads to formation of droplets of different height/density? That would render these results uncomparable. Another thing in question is the purpose of the panel (d). The data presented regards only droplets with the peptide and cannot be related to anything.

Minor remarks

The line 15.: perhaps this would be better: "acidic amino acid residues were involved". In general, authors use the term "amino acids" across the whole manuscript. However, the study if focused on amino acid residues in peptides, not free amino acids.

The line 16.: "based on the amino acid sequence of the original peptide"?

The line 18.: "highlighted the potential"?

Author Response

Point 1: The study lacks a valuable comparison with any antioxidant used in the food industry for oil preservation, and particularly with the most common one, which is alpha-tocopherol. Keeping in mind the limit on antioxidant amount in oil, mentioned in the line 62, it would be desirable to compare antioxidant potential of the studied peptides, or at least a representative one, with alpha-tocopherol within the same molar as well as mass concentration.

Response 1: We would like to thank you for the comments. It is our fault, because no more information was given in the original version of our manuscript. Actually, in our previous work (doi 10.1002/jsfa.9521), one of the most commonly antioxidant TBHQ in edible oil has been selected as a control. It was found that after 30 days of storage, the peroxide value of the peptide-added walnut oil was only 1.2% higher than that of the TBHQ-added walnut oil, and 30% lower than the peroxide value of the oil without any antioxidant. The information has been supplemented to the revised version (Please see line 40-44).

Point 2: Another question to be asked is why did the authors not examine the dipeptide Glu-Asp alone? There are reports demonstrating that different dipeptides have antioxidant porperties. Why not to check Glu-Asp in this study?

Response 2: Thank you for the comment. We have checked the antioxidant activity of the dipeptide Glu-Asp and found that Glu-Asp was more active when present in the peptide QITEGEDGGG than as a dipeptide. We have made a revision to Table 1, and modified the descriptions between line 224 and line 226.

Point 3: â‘ The last thing which bothers me is the figure 1 and the interpretation of the results. The panels (b) and (c) the oil droplets are much bigger than those in the panel (a). Are the graphics scaled properly? If so, is it possible that addition of the peptide alters the surface tension, which subsequently leads to formation of droplets of different height/density? That would render these results uncomparable. â‘¡Another thing in question is the purpose of the panel (d). The data presented regards only droplets with the peptide and cannot be related to anything.

Response 3: We really appreciate the comments.

â‘  The difference of oil droplets among Figure 1(a), Figure 1(b) and Figure 1(c) might depend on the angles which these photos were taken. We have replaced the old pictures of Figure 1(b) and Figure 1(c) with the new photos. Hopefully the new pictures will not cause misunderstanding this time. Please check Figure 1(b) and Figure 1(c).

â‘¡ Figure 1 (d) represents the summary of the results from Figure 1 (b) and Figure 1 (c). Normally, the fluorescence signals of peroxide and peptide in the oil droplets can be observed via confocal laser scanning microscopy, and the fluorescence signals can be expressed as their fluorescence intensity. Thus, data from Figure 1(d) showed that the intensity of fluorescence signal from the peroxide was reduced after the peptide was added to the walnut oil. We have modified our descriptions between line 151 and line 154. Hopefully this revised version will not produce misunderstanding.

Point 4: The line 15.: perhaps this would be better: "acidic amino acid residues were involved". In general, authors use the term "amino acids" across the whole manuscript. However, the study if focused on amino acid residues in peptides, not free amino acids.

Response 4: Thank you for the comment. We have added the “residues” to the revised manuscript.

Point 5: The line 16.: "based on the amino acid sequence of the original peptide"?

Response 5:Thank you for the comment. We have modified the sentence in the revised manuscript (Please see line 17).

Point 6:The line 18.: "highlighted the potential"?

Response 6:Thank you for the comment. We have made a revision to this word in the revised manuscript (Please see line 19).

Reviewer 2 Report

The present authors synthesized a decapeptide, QITEGEDGGG, and its derivatives and examined antioxidant activity using walnut oil, superoxide anion scavenging ability and linoleic acid oxidation system. However, objective and conclusion of this study seem not clear.

First, the reason why they used this peptide is not clear. Little description for the reason. Reader may assume that this peptide exerted antioxidant activity. But cannot understand it is strong compared to other peptide or not. The authors should explain why they used it such as it is acidic peptide with high antioxidant activity and so on. The objective of experiments such as fluorescence of lipid, Raman spectrum, GC-MS should be clarified. In addition, there are many confusing sentences and phrases as follow, which made difficult to evaluate this manuscript.

L57; What is C11-BODIPY 581/591?

L76; What is PBN?

L115; What is “Superoxide anion activity”?

L141; The intensity of the peptide was stronger than that of peroxide, indicating that the presence of peptide reduced the formation of peroxide in oil (Figure1(d)).  INTENSITY OF PEOTIDE!? I cannot understand.

165; “(b) ESR signal of the peptide during oil oxidation.”

What did you mean? ESR signal of peptide solution without lipid?

L209: “Lastly, the acidic amino acid combination Glu‑Asp was removed from inhibitor No.4 (QITEGGGG).”

??? Glu-Asp was removed from inhibitor No.0 QITEGEDGGG and synthesized No. 4 (QITEGGGG).

L210; “Compared with inhibitor No.0, the superoxide anion radical scavenging activity and the inhibition activity of linoleic acid oxidation of inhibitor No.1 was significantly decreased”

The No 1 may be No. 4?

L230: “In this work, we investigated the effect of acidic amino acids of inhibition of walnut oil oxidation using a peptide lacking hydrophobic amino acids but rich in acidic amino acids.”

Did you mean that “a peptide lacking hydrophobic amino acids but rich in acidic amino acids” is QEGEDGGG (No. 3)? But this peptide was evaluated only by superoxide anion radical scavenge ability and linoleic acid oxidation assay. Not by walnut oil oxidation. If you meant QITEGEDGGG, it consisted of hydrophobic amino acid (Ile).

L245: “Raman shifts of the un‑oxidized walnut oil, OWO, OPO and the peptide proved that the lipid oxidation interfered by acidic amino acids of peptide.” The Raman shift experiment used only one peptide. How did you conclude that acidic amino acids interfered oxidation?

Author Response

Point 1: The reason why they used this peptide is not clear. Little description for the reason. Reader may assume that this peptide exerted antioxidant activity. But cannot understand it is strong compared to other peptide or not. The authors should explain why they used it such as it is acidic peptide with high antioxidant activity and so on. The objective of experiments such as fluorescence of lipid, Raman spectrum, GC-MS should be clarified. In addition, there are many confusing sentences and phrases as follow, which made difficult to evaluate this manuscript.

Response 1: We would like to thank reviewer’s comments very much.

â‘ Regarding the problem on “Little description for the reason”, we have modified our descriptions in the revised manuscript. The missing information have been supplemented to the revised version (Please see line 40-44; line 48-50).

â‘¡ Regarding the question on “the objective of experiments such as fluorescence of lipid, Raman spectrum, GC-MS”, we have been supplemented the missing information to the revised manuscript (Please see line 69-70; line 81-83; line 94-95; line 77-79; line 104-105).

â‘¢ Regarding the questions on “many confusing sentences and phrases”, we are so sorry for this. We have asked a professional proof-reading company to check our revised manuscript again. Hopefully we could minimize the errors as much as possible this time.

Point 2: L57, What is C11-BODIPY 581/591?

Response 2: Thank you for the comment. We are sorry for this. C11-BODIPY 581/591 is a high quality fluorescent probe for labelling peroxides. The information of C11-BODIPY 581/591 has been added to the revised manuscript (Please see line 73-74).

Point 3: L76, What is PBN?

Response 3: Thank you for the comment. We are sorry for this mistake. PBN is an ESR spin trapper to capture the lipid-derived carbon- centered radicals in oxidized walnut oil. The information of PBN has been added to the revised manuscript (Please see line 85-86).

Point 4: L115, What is “Superoxide anion activity”?

Response 4: Thank you for this comment. We have corrected this phrase in the revised manuscript (Please see line 126-127).

Point 5: L141,The intensity of the peptide was stronger than that of peroxide, indicating that the presence of peptide reduced the formation of peroxide in oil (Figure1(d)). INTENSITY OF PEPTIDE!? I cannot understand.

Response 5: We would like to thank you for the comments. We are sorry to make you confused. We have revised this part as follows. 

â‘  “Data from Figure1(d) showed that the intensity of the peptide fluorescence signal was stronger than that of the peroxide. It means that the presence of the peptide retards walnut oil to form more peroxides”. Hopefully the revised version will produce less misunderstanding (Please see line 151-154).

â‘¡Regarding this problem on “INTENSITY OF PEOTIDE”, we have corrected our descriptions in the whole text. The “INTENSITY OF PEPTIDE” has been changed to “the intensity of fluorescence signal of peptide”.

Point 6: L165,“(b) ESR signal of the peptide during oil oxidation.” What did you mean? ESR signal of peptide solution without lipid?

Response 6: Thank you for this comment. “ESR signal of the peptide during oil oxidation” means “ESR signal of heated-treated peptide solution without oil”. We have corrected the phrase in the revised manuscript (Please see line 175-176).

Point 7: L209: “Lastly, the acidic amino acid combination Glu Asp was removed from inhibitor No.4 (QITEGGGG).” Glu-Asp was removed from inhibitor No.0 QITEGEDGGG and synthesized No. 4 (QITEGGGG).

Response 7: We would like to thank you for the comments. We have corrected the descriptions in the revised manuscript (Please see line 220-221).

Point 8: L210; “Compared with inhibitor No.0, the superoxide anion radical scavenging activity and the inhibition activity of linoleic acid oxidation of inhibitor No.1 was significantly decreased” The No 1 may be No. 4?

Response 8: Thank you and we are very sorry for this mistake. We have corrected this error in the revised manuscript (Please see line 221; line 223).

Point 9: L230: “In this work, we investigated the effect of acidic amino acids of inhibition of walnut oil oxidation using a peptide lacking hydrophobic amino acids but rich in acidic amino acids.” Did you mean that “a peptide lacking hydrophobic amino acids but rich in acidic amino acids” is QEGEDGGG (No. 3)? But this peptide was evaluated only by superoxide anion radical scavenge ability and linoleic acid oxidation assay. Not by walnut oil oxidation. If you meant QITEGEDGGG, it consisted of hydrophobic amino acid (Ile).

Response 9: We really appreciate you for the comments. Indeed, this sequence QITEGEDGGG presented in our work contains one hydrophobic amino acid (Ile). Thus, we have deleted the description on “hydrophobic amino acids”, and only addressed “acidic amino acids”. Please see the revised version (line 244-245).

Point 10: L245: “Raman shifts of the un-oxidized walnut oil, OWO, OPO and the peptide proved that the lipid oxidation interfered by acidic amino acids of peptide.” The Raman shift experiment used only one peptide. How did you conclude that acidic amino acids interfered oxidation?

Response 10: Thank you for the comments. Firstly, we discussed the mechanism of action of the peptide QITEGEDGGG in delaying walnut oil oxidation by Raman spectroscopy. Next, acidic amino acid residues were found to be involved in the inhibition of oil oxidation. Finally, systematically screening of amino acid residues in the peptide were done to determine the possible role of acidic amino acid residues. Thus, it was speculated that acidic amino acids of the peptide might be important in the reduction of lipid oxidation. We have modified the expression in the revised manuscript (Please see line 259-261).

Round 2

Reviewer 1 Report

Dear Authors, thank you for your answer. I feel that most of my remarks have been addressed appropriately.

However, there's still a doubt related to to figure 1. I believe it still needs some rephrasing/clarification.

You write in the paper:

"The fluorescence signal of peroxide in oil droplets of walnut oil which underwent oxidation is displayed in Figure 1(a).

Figure 1(b) is the fluorescence signal of peroxide in oil with the peptide added (peptide-oil) and

Figure 1(c) presents the fluorescence signal of the peptide in the peptide-oil mix.

Figure 1(d) represents the summary of the results from Figure 1(b) and Figure 1(c).

The fluorescence signals of peroxide and peptide in the oil droplets can be expressed as their fluorescence intensity. Thus, data from Figure 1(d) shows that the intensity of the fluorescence signal from the peroxide was reduced after the peptide was added to the walnut oil."

And as a response: "Figure 1 (d) represents the summary of the results from Figure 1 (b) and Figure 1 (c). Normally, the fluorescence signals of peroxide and peptide in the oil droplets can be observed via confocal laser scanning microscopy, and the fluorescence signals can be expressed as their fluorescence intensity. Thus, data from Figure 1(d) showed that the intensity of fluorescence signal from the peroxide was reduced after the peptide was added to the walnut oil. We have modified our descriptions between line 151 and line 154. Hopefully this revised version will not produce misunderstanding."

The panels (b) and (c) describe only the oil-peptide mixture, as I presume exactly the same one. One shows the fluorescence intensity of peroxide, the other of the peptide. You can't assess based on that the reduction in peroxide amount that can be attributed to the peptide presence. You would need to compare it to a mixture without a peptide (panel a), or, even better, with the peptide added at the end of incubation compared to a mixture with a peptide added at the beginning of incubation. I guess the panel (d) shows now completely unrelated fluorescence intensities of the peptide and peroxide in three chosen droplets from the peptide-oil mixture and nothing else. There's no comparison with the peroxide level in mixtures without the peptide, no reference, no control. In my humble opinion, based on what's visible in the panels (b) and (c), you can't say the peroxide level was reduced, reduced when compared to what?

If you mean something else, please rephrase the description so it can inform about your intentions properly.

Author Response

Point 1:  There's still a doubt related to figure 1. I believe it still needs some rephrasing/clarification. The panels (b) and (c) describe only the oil-peptide mixture, as I presume exactly the same one. One shows the fluorescence intensity of peroxide, the other of the peptide. You can't assess based on that the reduction in peroxide amount that can be attributed to the peptide presence. You would need to compare it to a mixture without a peptide (panel a), or, even better, with the peptide added at the end of incubation compared to a mixture with a peptide added at the beginning of incubation. I guess the panel (d) shows now completely unrelated fluorescence intensities of the peptide and peroxide in three chosen droplets from the peptide-oil mixture and nothing else. There's no comparison with the peroxide level in mixtures without the peptide, no reference, no control. In my humble opinion, based on what's visible in the panels (b) and (c), you can't say the peroxide level was reduced, reduced when compared to what?

Response 1: We really appreciate you for the comments. Indeed, here is a lack of comparison of peroxide level before and after peptide addition in Figure 1(d). Thus, we added the data of control, that is, the mean value of fluorescence intensity of peroxide in the oil without a peptide to the Figure 1(d). And we have modified our descriptions between line 152 and line 161. Hopefully this revised version will not produce misunderstanding (Please see Figure 1(d)).
